# The Sustainability of Keeping Birds as Pets: Should Any Be Kept?

**DOI:** 10.3390/ani11020582

**Published:** 2021-02-23

**Authors:** Shawn Peng, Donald M. Broom

**Affiliations:** 1Taipei Zoo, No.30, Sec.2, Xinguang Road, Taipei 116, Taiwan; 2Department of Veterinary Medicine and St Catharine’s College, University of Cambridge, Madingley Road, Cambridge CB3 0ES, UK; dmb16@cam.ac.uk

**Keywords:** pet birds, biodiversity, welfare, ethics

## Abstract

**Simple Summary:**

The trade in birds for pet trade is harming wild bird populations and bird welfare. Inadequate housing of many pet birds results in stereotypies and other indicators of poor welfare in the birds that are currently widespread. Some pet birds were taken from the wild or bred in poor conditions, while others had nutritional, health, and behavioral problems resulting from inadequate living conditions and incorrect husbandry by the bird owners. As a consequence, it is not ethically right to keep the majority of the birds that are at present kept as pets. If birds are to be continued as a companion animal for people, then more effort should be made by pet shop owners and veterinarians to supply literature to prospective owners highlighting not only the proper care for the species of bird but also its needs and requirements, so that bird owners can do their utmost to meet them. Owners do not comply with laws requiring duty of care, unless they obtain and act on such information and also have knowledge of how to provide good nutrition and minimize the risk of disease. New laws are needed to prohibit taking birds from the wild and keeping birds in conditions that do not meet their needs.

**Abstract:**

We describe a wide range of unethical and unsustainable practices inherent to the trading and keeping of pet birds. At present, biodiversity and wild bird populations are being greatly harmed and many individual birds have poor welfare. Wild-caught birds should not be sold to the public as pets, or to breeding establishments for several reasons, including because 75–90% of wild-caught birds die before the point of sale and taking birds from the wild has negative effects on biodiversity. The housing provided for pet birds should meet the needs of birds of that species and allow good welfare, for example there should be no small cages but aviaries with space for each bird to exercise adequately, and social birds should be kept in social groups. At present, inadequate housing of many pet birds results in stereotypies and other indicators of poor welfare in birds. Owners should have knowledge of how to provide good nutrition and minimize the risk of disease. Unless these changes are made, keeping birds as pets should not be permitted. New laws are needed to prohibit taking birds from the wild and ensure captive pet birds in conditions that do meet their needs.

## 1. Introduction

The sustainability of all human interactions with other animal species and the advisability and morality of taking animals from the wild is now being questioned [1,2,3,4]. A system or procedure is sustainable if it is acceptable now and if its expected future effects are acceptable, in particular in relation to resource availability, consequences of functioning, and morality of action [5]. Some aspects of obtaining and selling birds and keeping them in captivity as pets are not acceptable to a high proportion of the public, in particular because of biodiversity loss, impact on conservation policies, and poor welfare of the birds —starting from capture, to shipping, and restraining birds in captivity. The consequences of the trade in pet birds, discussed here, are important to conservation biologists understanding the impact of pet bird market versus habitat loss, and veterinarians’ response to rescuing smuggling birds, as well as to the global public awareness of consequences of keeping pet birds.

In the United States, pet birds are reported to be the fourth most popular companion animal to keep as a pet, after fish, cats, and dogs [6,7], while in the EU, ‘ornamental’ birds were the third commonest pet when fish were not counted individually [8,9,10], explaining that unlike other pets such as cats and dogs, birds are not usually considered to be domesticated animals, even when they are a captive bred species. Domestication is the process, occurring over generations, through which a population of animals becomes adapted to man and to a captive environment through some combination of genetic changes and an environmentally induced developmental events [11]. Many birds in captivity are only one or two generations removed from the wild but even in birds bred for more generations like Canaries *(Serinus canaria)*, Budgerigars *(Melopsittacus undulatus)*, Zebra Finches *(Taeniopygia guttata)*, Lovebirds (*Agapornis* sp), and Cockatiels *(Nymphicus hollandicus)*, their behavior and physiology differs little from that of wild individuals [12]. Furthermore, unlike cats and dogs, captive bred birds are mostly physically identical to their wild counterparts, the only exceptions being the few species bred to express color mutations or hybridization [9,10,13].

EFSA reported that birds from 17 different orders were taken from the wild and exported for pet trade from 43 countries [14]. The taking of birds from the wild has substantial negative impacts on conservation and biodiversity worldwide. Ribeiro et al. estimated that one-third of all endangered wild bird species, totalling more than 400 species, are threatened because of the pet trade [15].

IUCN categories now have to change much faster than usual, to reflect the actual situation—critically endangered, endangered or vulnerable [16], while low public awareness results in many significant differences in conservation status between the IUCN Red List, CITES index, and the species given protection under National law. Hence, there are several ethical and scientific arguments concerning the keeping of birds as pets.

## 2. Ethical Issues Associated with Keeping Pet Birds

### 2.1. Principles

When considering the ethical issues of keeping pet birds, three principles of ethics can be considered [17]—(i) the welfare of the individuals involved, (ii) the autonomy and freedom of choice for each individual, and (iii) justice for each action involving respect for the principle of fairness to all. Using these principles, Mepham proposed that they should be applied to—the subject of animals, in this case the birds, the people involved in the trade, the consumers, i.e., the pet keepers, and the environment, including that from which the subjects originated [18]. The welfare of wild and domestic animals and conservation of the environment are major factors in ethical decisions about trade for the public, for example in the EU, as is indicated by the Eurobarometer surveys in relation to farm animal welfare and the killing of seals for producing sealskins [19,20]. While the welfare of the birds is the major topic considered in this paper, other ethical issues would also be considered briefly.

### 2.2. Positive Effects on People

Pet birds can be a major source of comfort to some human owners [21]. As summarized by [6], the owners might show improvements in mental and physical health. Some of these effects, such as reduced blood pressure and reduced depression when spending time with a pet bird, are objectively measurable, while the evidence for others is owner reports—a source of data susceptible to bias. Owners like to be able to decide to keep pet birds and many do not wish to be restricted by regulations, such as the limitations on what species can be kept, associated with CITES compliance or other national laws, or laws preventing the keeping of birds in certain conditions. Some of those who keep pet birds, do it in order to impress other people with the exotic animals in their home and might not do so if they had more information about the consequences [22].

Those who sell birds directly to potential owners or those who breed, transport, and market birds to retailers, benefit from their trade. For many species, the people who catch wild birds in order to sell them are often relatively poor, and consider the wild as a legitimate and traditional source of revenue that is beneficial to them. However, many people living in or close to wild environments do not want wild areas or populations to be destroyed.

### 2.3. Effects on Environment Where Birds Are Kept

The environment in places where birds are in captivity is not greatly affected by the presence of the birds, except for disease transmission risks, as described in the next paragraph, and a limited amount of pollution from bird excrement and wasted food. Normally, this can be adequately managed. Further environmental risks occur if captive birds escape. In addition to possible transmission of disease to other animals, the escaped birds sometimes establish breeding populations that have negative effects on some native species [23]. Examples of this include Rose-Ringed Parakeets (*Psittacula krameria)* and Monk Parakeets (*Myiopsitta monachus)* in European countries, where they are considered to be a major cause of biodiversity loss [24]. A different kind of environmental impact of living with captive birds is that those people who live with pets during childhood are usually more sympathetic to captive and wild animals later in life and more likely to support conservation activities [25].

An environmental impact of captive birds is that they are often a reservoir of disease that might affect farmed birds or humans. When the birds, their accommodation or their excrement come into direct contact with people or other pets, diseases such as ornithosis, psittacosis, or influenza can be transmitted. Examples of captive birds presenting a health risk to their owners include the following. Reference [26] report on a case of asthma in a 28-year-old woman resulting from the ownership of two Orange-Ccheeked Waxbills (*Estrilda melpoda*). Vereda described a case of rhinoconjunctivitis and asthma that was caused by pet Lovebirds [27]. Additionally, Holst and Kohlmeier found there to be an increased risk of lung cancer with ownership of a pet bird [28,29]. Other diseases that cause problems for farmed birds, such as Newcastle disease, could be transmitted to them when wild-caught birds are subsequently kept in captivity without adequate quarantine and other precautions. The risks of transmission of diseases from wild-caught birds to humans or domestic animals are described by [14]. One conclusion was that the zoonotic potential is high for chlamydia originating from Psittacines, Domestic Ducks, Geese, and Turkeys and that this is a very serious risk. Other conclusions concerned the risks of captive birds as a source of influenza in humans or poultry and Newcastle disease in poultry.

### 2.4. Autonomy of Birds

The autonomy of the birds that are captured, bred in captivity, and kept as pets is very low. The birds have no control over what happens to them. As mentioned in relation to welfare below, they do not choose to be captured and very few of them choose to remain in captivity. Do these birds gain anything from such captivity? The most successful bird in the world in terms of numbers is the Domestic Fowl (*Gallus gallus)* [11]. Some birds kept as pets are much more numerous than they would be if they were living only in the wild. However, while this is a benefit to the species, it is not a benefit to the captive individuals.

### 2.5. Effects on Biodiversity

The effect that the trade in birds to be kept as pets has on biodiversity was and continues to be great. DeLong defines biodiversity as the variety within and among living organisms, biotic communities, and biotic processes, whether naturally occurring or modified by humans [30]. Some bird species were made extinct as a result of being captured for pet trade and for many others their population size is greatly reduced. Some of the capture methods directly affect other species because capture of the target species involves killing of non-target species. Other species are adversely affected because of disturbance associated with capture attempts, removal of a food source, or other disruption caused by removal of birds that provided some resource for the animal species. The removal of a species can also result in increased growth of plants or animals that would formerly have been consumed by the species removed. The removal from an area of species, such as large, colorful parrots can also alter human enjoyment and behavior, with various consequences for the species still present. Trade in birds caught for the pet industry can involve widespread law-breaking, sometimes by violent criminals [31], and the presence of these people in a human community can have negative effects on the humans and the wildlife. Although the threats to wild bird populations in Asia are mostly due to habitat loss, wildlife trade is considered to be another major cause in the region. Nijman reported on 1 million CITES-listed birds, almost all wild-caught, which are known to have been taken from South East Asia in a ten-year period [32]. This is also true for many other parts of the world. In markets in many countries, traders openly sell illegally trapped and protected birds without being punished [33]. There is little public awareness of this, and despite widespread appreciation for wild birds, many are still kept in the house without consideration of the impact that this practice has on wild bird populations and biodiversity in general.

## 3. Animal Welfare

### 3.1. Welfare and Needs

One of the main issues for the public in relation to the keeping of birds in captivity as pets, is the consideration of the animal’s welfare. Broom defines welfare as “the state of an individual as regards its attempts to cope with its environment” [34,35]. Broom and Fraser explain that animals have many coping mechanisms, including positive and negative feelings, and a wide range of needs that are a consequence of the many functional systems that make life possible [11]. It is important to meet these needs in order to ensure good welfare of captive animals. The concepts of welfare and needs have exactly the same meanings for all animals, including humans. General guides about what to provide for captive animals are the five freedoms or the five or twelve domains but current research provides more detailed information about the needs of each species. As a basis for their analysis of welfare, EFSA listed that birds need to—breathe, rest and sleep, exercise, avoid fear, drink and feed, have access to an appropriate hiding or resting place, explore, have social contact, minimize disease, preen, thermoregulate, avoid harmful chemical agents, and avoid pain [14]. They pointed out that the needs of birds vary according to their way of life and biological adaptations to their environment and that if needs are not met, the welfare of the animal becomes poor, either slowly or rapidly. The link between poor welfare and susceptibility to disease in captive birds was emphasized, together with behavioral and physiological responses that are a response to being disturbed by handling or an inadequate environment. These needs were used in categorizing hazards for a welfare risk analysis.

### 3.2. Breathing

While birds used as companions in people’s homes normally have adequate clean air to breathe, during transport prior to sale, birds might be in very small cages or boxes with inadequate air supply, or might be exposed to noxious or toxic fumes, for example from vehicle exhaust. Insufficient air or poor air quality is one of the causes of death during transport, especially if the birds are crammed into a small container, as might occur when trading birds is illegal. Mortality figures for captive wild birds during handling and transport are much much higher than those for mortality during transport of any other animals used by humans [14]. Any difficulty in breathing causes fear in birds, as it does in other air-breathing animals, and fear causes very poor welfare additional to the substantial direct effects of the deprivation.

### 3.3. Rest and Sleep

Holding conditions after capture of wild birds and transport of all captive birds might involve much disturbance and movement that prevents normal rest and sleep. Some owners of pet birds keep them in conditions where rest and sleep are difficult. Prolonged deprivation of rest and sleep has negative effects on welfare.

### 3.4. Exercise

Exercise is not a luxury but is a necessity for maintaining effective body and brain function. For example, hens kept in battery cages have insufficient exercise, with the consequence that they have osteoporosis and are likely to break their bones when handled [36]. Captive animals, elderly humans, and astronauts in zero gravity are vulnerable to muscle wastage and osteoporosis unless their muscles and bones have load-bearing activity [11]. Birds in small cages are often unable to fly, or to fly far enough for the exercise to be sufficient [37]. While there might be variation amongst bird species in what amount of flight is sufficient, a guide suggested for all animals by [3] was that each animal should be able to show normal locomotion for at least five seconds. A bird that flies at 30 km per hour would fly 42 m in five seconds, therefore, if it can easily fly in a circle, it would need a circular cage of at least 4.8 m in diameter. Exercise is not needed continuously but at intervals during each day, perhaps 5–20 times more exercise is needed by smaller birds with higher metabolic rates.

Flight requires fully functional wings, and discrepancies in wing function are very disturbing to birds. Athan studied “wing-clipping”, which is commonly done to keep pet birds from flying [38]. This can be done either to prevent the bird from flying away and escaping or, in an attempt to keep the bird safe, as the home where they live is dangerous for free-flying birds. However, wing-clipping should be avoided, except in rare situations where it improves bird welfare, and the process needs to be carried out correctly with the trim being functional and appropriate to the species of bird [39]. For example, asymmetric trimming can lead to major welfare problems. A study using electromyography [40] indicated that restricting wild Great Mynahs (*Acridotheres grandis*) in a small pet cage resulted in pectoralis muscle quantity decrease. A preference test in the same study revealed that the birds preferred cages with more space or the same space volume but with a greater height. 

Insufficient space for normal functioning can also be caused by overcrowding in the cages. When too many birds of any species are kept in one cage or aviary, the result is discomfort and increased aggression. For example, Maat et al. carried out studies on the behavioral and physiological effects of population density in enclosures housing Zebra Finches (*Taeniopygia guttata*) [41]. Zebra finches are extremely social birds living in stable colonies and they pair for life [42,43,44]. Maat et al. found that aggression was higher in high density enclosures and concluded that low density populations were better for welfare than high density conditions [41]. 

### 3.5. Avoid Fear

Wild birds are very fearful of being confined or being close to what they perceive as a dangerous predator, for example a human. The extreme reaction associated with fear is one of the reasons why the mortality rate of birds taken from the wild and put into cages is very high. Other reasons include difficulty in adapting to different food and water sources and disease, often associated with immunosuppression caused by the stress of captivity. When small birds were captured in the wild for sale as pets, 90% were dead before reaching an owner and even for high-value birds such as the larger parrots, 75% died before retail sale [14]. As a consequence of knowing this, most people do not accept the capture of wild birds for sale as a pet and many countries have banned it.

Fear of humans is a major problem for many companion birds, even if captive bred. However, in some species, and after appropriate early experience including human interactions, such fear can be reduced to a low level. Birds are fearful of predators, including domestic cats and dogs, so this should always be considered when birds are kept as pets. Fear and distress can also result from incorrectly carried out veterinary and routine procedures and learning from previous experience of frightening situations can result in birds predicting frightening events and their welfare being poor, when such situations recur. 

Feather damage resulting from feather-pecking can be associated with fear of the pain of being pecked by other birds, for example in caged hens [45,46], it is considered that the fearful birds are the ones doing the pecking and that the more the birds peck other birds, the more fearful they are. Fear in captivity was also documented in quail, which, due to their poor enclosures, carry out less dust-bathing activity and this in turn leads to increased susceptibility to fear [47].

### 3.6. Drinking and Feeding

Captive birds might react to frightening and stressful situations, such as capture, transport, potential non-human predators, or human handling, by avoiding drinking and feeding. They might also be harmed if they are not given appropriate food by their human owners. The continued development of carefully compounded bird food available to pet owners reduced the prevalence of former common diseases, which were associated with pet birds including protein deficiency, rickets, and hypothyroidism [10,48]. However, despite the fact that commercially available bird seed is nutritionally sound for birds, obesity is a common issue with many pet birds because they get insufficient exercise and are over-fed.

Harrison highlights some guidelines for the feeding of pet birds in order to ensure adequate nutrition [49]. These guidelines include the buying of fresh food every 4–6 weeks, correctly storing food in a cool and dry environment, giving the bird fresh food daily, and feeding the correct type and amount of supplements with the standard diet. The main issue is that many bird owners might not follow these guidelines. Indeed, Harrison points out that rather than feeding fresh food daily, it is common for bird owners to just “blow off” husks and top up the seeds in the birds’ bowl [49]. Diseases that can occur in pet birds as a result of being given an incorrect diet are described below and by [50].

#### 3.6.1. Amino Acid and Protein Problems

Hagan analyzed a number of seed-based diets that revealed that the proteins lacked sufficient quantities of amino acids—lysine, methionine, and threonine [51]. Amino acid deficiencies are important in seed-eating bird species, especially for passerine and psittacine species that would use animal-based protein sources in the wild. 

#### 3.6.2. Fat and Energy Problems

Fat and energy requirements for birds vary according to natural diet and seasonal activities. Excess lipid concentrations cause health problems such as obesity, which is common in adult parakeets, cockatiels, and Amazon parrots. Inactivity and high dietary fat concentrations are the main causes. Another related health issue is hepatic lipidosis, which is prevalent in Amazon parrots [48,52,53].

#### 3.6.3. Vitamin Problems

Seed-based diets for captive birds often cause vitamin A deficiency; signs of which are nasal discharge, poor growth, and poor feather formation [50,54]. This deficiency can also cause bumblefoot and inadequacies in growth, reproduction, vision, and the immune system. Vitamin A is obtained in the wild from leaves, shoots, and fruits. Other vitamin deficiencies, common in captive birds, are in vitamin D, which might cause rickets, osteodystrophy, hypocalcaemic tetany, and bone malformations, and in Vitamin E, which can cause weak legs, digestive problems, and edema in captive Cockatiels [55] and poultry [54].

#### 3.6.4. Mineral Problems

Calcium, phosphorus, iodine, iron, zinc, sodium, and chloride deficiencies are of particular concern in pet birds [50]. Calcium supplements are important for laying birds and can be obtained from cuttlefish bone, oyster shell, or broccoli, and dandelion leaves. Iodine deficiency is common in some parakeets [56,57].

### 3.7. Access to a Hiding and Resting Place

A high proportion of birds spend much of their time hiding from potential danger. This is an important component of their biology and prevention of hiding causes fear and stress. Many captive birds have nowhere to hide from humans, other potential predators or events perceived as dangerous. Birds also need places to rest where they are comfortable and feel safe. Most birds used as companions are given a perch but some are not. Poultry such as the Domestic Fowl choose to rest on a perch that is elevated above ground level. Absence of a suitable perch and a hiding place is one of the major causes of abnormal behavior in captive birds. Several behavioral changes associated with captivity, such as biting and other aggression, loud vocalizations, psychogenic water and food consumption, regurgitation, escape attempts, feather-picking, stereotypies, and suppression of reproduction are a consequence of the failure of the environment provided to meet such needs of the bird [58,59,60].

### 3.8. Explore and Have Social Contact

Exploration has a function in relation to danger avoidance, finding adequate resting places, finding food, and social behavior. Birds in cages have little possibility for exploration. All bird species kept as human companions have a high level of cognitive ability, and research on parrots and corvids shows that these animals have a level of sophistication of brain function superior to that of most monkeys [5]. Most species kept as companion animals are social, and for social species of birds, being deprived of social interaction with members of their own species has extremely negative effects on their welfare. Humans are not adequate substitutes for their own species, even if human company can confer benefits for some individuals that have had appropriate previous experience.

One of the key causes of fear and distress for social birds in captivity is separation from family and mates. Zebra finches mate for life [43,44] and separation of a bonded pair of zebra finches was shown to cause stress in captive birds, indicated by changes in behavior and an increase in corticosterone levels [61,62]. Long-term individual housing of pet birds is the greatest cause of poor welfare in these birds.

Restriction of normal behaviors because the birds are in a caged environment can lead to modified or abnormal behaviors in the animal. Feather-plucking is one of the behavioral abnormalities that occurs in 10% of captive parrots [63] because of restricted opportunities for normal behavior [64,65]. Lumeij and Hommers concluded that feather-plucking occurs because the captive birds have little opportunity to forage [66]. However, parrots kept in cages are often deprived of companions, exercise, and overall complexity of experience, as well as foraging opportunities. Reference [67] measured telomere length, which is shorter in stressed people and found that grey parrots living in isolation had shorter telomeres than those in pairs. Reference [13] concluded that parrots are not suitable to be companion animals because of the widespread indications of poor welfare in parrots kept at home.

Stereotypies and other abnormal behaviors shown by captive birds are widely reported in parrots and cockatoos, but can occur in a wide range of species. Reference [68] carried out studies on the behavior of captive European starlings, which are the most common passerine species used in laboratory research [69]. These starlings are often individually housed, which is a major risk factor for the development of stereotypic behaviors in many species of bird [70,71,72]. One of the common stereotypies shown by captive starlings is somersaulting [73]. Reference [68] states that this behavior comes about because of repeated, failed escape attempts. The behavior occurs repeatedly at the same location within the bird’s cage. Attempts to escape often become stereotypies, and birds that spend a lot of time on the cage walls and ceilings, usually make escape attempts, and are therefore frustrated and have poor welfare [74].

The rearing method is also a factor that affects the behavior of parrots in captivity. Reference [75] studied the effects of hand-rearing on the behavior of African gray parrots. Reference [75] explain that African gray parrots are one of the most commonly kept companion animals across the world, and that hand-rearing by humans was increasingly carried out during the last 25 years. One of the problems with hand-rearing, as highlighted by [75], was that hand-reared parrots choose a specific human as a partner and this leads to frustration in the birds as the human bonds do not fully satisfy the parrots’ social requirements. As a result, hand-reared parrots might develop attention seeking behavioral disorders, including increased aggressiveness, feather-picking, stereotypies, and even abnormal sexual behaviors. Reference [75] confirmed that, in gray parrots, the breeding method has an effect on behavior and hand-rearing leads to a higher incidence of behavioral disorders. 

Treatments sometimes used for behavioral disorders are psychoactive drugs. These might be used to treat feather-chewing, self-mutilation, and anxiety [76]. Examples of such psychoactive drugs include carbazemine, which is an anticonvulsant, diazepam which is used for acute stress and for appetite stimulation, and haloperidol which is an antipsychotic. However, there are health risks involved with the use of these drugs, for example, carbamazepine can cause suppression of the bone marrow and hepatotoxicity [77]. Opioid antagonists are used to block behavioral problems, such as feather-plucking and soft-tissue mutilation, but have the possible side effect of increasing anxiety and gastrointestinal problems, such as abdominal cramping, nausea, vomiting, and constipation [76].

### 3.9. Minimise Disease

All disease causes some degree of poor welfare and this can be extreme in some pet birds. Van Der [78] explains that mycobacterial diseases are a significant cause of mortality in pet birds, the most common causes being *Mycobacterium avium* and *M. genavensa.* Mycobacteriosis is commonly contracted through contaminated substratum and contaminated water. When the bird accommodation is not cleaned properly, or does not have its water changed at regular intervals, there is heightened risk of mycobacterial diseases.

Van Der Heyden carried out a study on diseases of the respiratory tract in companion birds [79]. He also points out that a key cause of the respiratory tract disease rhinitis, comes from foreign bodies found in the bird’s enclosure. These foreign bodies range from small particles of food, such as powder from pellet feed, to wood chippings, to even items put in as a form of environmental enrichment, such as pieces of paper put inside the enclosure for the parrot to tear apart. Small foreign bodies are inhaled by the bird, become lodged in the nasal cavity, and lead to rhinitis. Bauck et al. and Morrisey state that rhinoliths and proliferative nasal granulomas occur in many Psittacine species but are frequent in African Gray Parrots (*Psittacus erithacus*) [53,79]. Other contributing factors to this ailment are malnutrition, especially vitamin A deficiency, poor air quality and ventilation, inadequate humidity, and lack of bathing facilities such as baths or showers. Rhinitis can also be fungal-based with African Gray Parrots, Amazon Parrots (*Amazona* spp.), and Cockatoos being particularly susceptible [49]. Other important diseases that affect captive bird welfare include sinusitis, which is common in pet birds due to the dorsal location of openings into the birds conchae, the causes being primarily bacterial and nutritional [79]. This is also seen in choana which was reported in young African Gray Parrots and adult Umbrella Cockatoos (*Cacatua alba*) [80]. In many parrots, trachea disorders like aspergillosis occur and are caused by poor hygiene, excessive humidity, excessive darkness in the enclosure, and dusty conditions, which allow the development of both fungal and bacterial tracheitis. Fillipich studied neoplasia, a disease affecting the urogenital system, especially in psittacine species [81]. In 15,000 birds in the USA, 9.95% were diagnosed with neoplasia [82]; 15.8% of Budgerigars (*Melopsittacus undulatus*), 1.4% of Galliformes, and 0.89% of Anseriformes (0.89%) had neoplasia in a study by Ratcliffe (1933). In addition to poor welfare in captive birds caused by disease, despite advice available to bird owners, humans contract disease from their pets, e.g., Salmonella infections [83].

Parasitic infection is also a major health concern for pet birds, especially those housed in an aviary. Johnson-Delaney explains that many avian species in captivity might be infected with the parasites *Cryptosporidium* and *Giardia* [84]. Another parasitic risk for birds is *Caryspora*, which infects the intestinal tract of birds [85]. Oocysts of *Caryspora* are produced in enterocytes of birds and carnivorous reptiles, but rodents might also serve as a host. Two other parasites known to affect pet birds are *Entamoeba histolytica* and *E. polecki*, which cause amoebic dysentry in humans and other animals [85].

### 3.10. Preening and Other Cleaning Behaviour

Flying animals have a key need to keep their wings clean. Hence having space to carry out normal preening behavior is very important for captive birds. When Jungle Fowl (*Gallus gallus spadiceus)* are in a captive environment with a wire floor or lacking manipulable material, they are not able to show another cleaning behavior—dust-bathing. Alternatively, dust-bathing might be abnormal or replaced by more pecking behavior that is associated with fear [86].

### 3.11. Thermoregulate and Avoid Harmful Chemical Agents

Most birds kept as companion animals are able to thermoregulate and to avoid toxic materials in their environment but, if not, their welfare can be very poor.

### 3.12. Avoid Pain

If the captive environment for birds is too small or inappropriately designed, there can be a risk of injury to the birds. For example, a natural behavior of Japanese Quail (*Coturnix japonica*) when disturbed, is to fly vertically into the air and then land again [87]. When this behavior is attempted in captivity, a low cage roof can cause severe head and neck injuries and even fatalities [88]. This behavioral problem can be prevented if the bird has adequate cover but there is always a risk that quail owners might not be aware of the behavior and, as a result, captive quails could suffer injuries. Cages and aviaries can sometimes have design faults that cause painful injuries, and both disease and aggression by other birds can cause pain.

### 3.13. Tests of Captive Bird Welfare

Animal welfare scientists can use a wide range of measures of behavior, physiology, pathology, etc. to assess the welfare of captive birds and the adequacy of the housing conditions and management methods. While more research on pet bird welfare would be useful, there is already a lot of information that can be used when deciding what should be permitted and what should not. Life expectancy and ability to reproduce are indicators of the welfare of each species and housing and management systems. If birds do not live as long as they could or do not reproduce as well as they could, either the species should not be kept or the housing and management method should not be used. 

The preferences of the subject animal are now widely used in animal welfare research. If the welfare of a companion animal is good in a particular housing system, most individuals should stay in the environment or return to it, even when given the opportunity to leave. If the pet owner opens the cage door, does the bird leave? If it leaves, does it come back, for example at feeding time, as most dogs, cats, and farm animals would do? Some owners of captive birds can confidently say that their birds do choose to stay with them. Most owners would not expect that their pets would stay, so it must be concluded that the welfare of these birds is not good. When the concept of justice, mentioned above, is considered, is it just to keep an individual that does not want to be kept in captivity?

## 4. Law

### 4.1. Legislation about Captive Bird Welfare

Bird in zoos, including bird parks open to the public, receive attention and welfare monitoring as a consequence of each country’s zoo legislation. The laws vary but might cover factors affecting welfare, such as pinioning, flight restraint, or cage size [40]. Regional zoo associations such as AZA (American Association of Zoos and Aquariums), EAZA (European Association of Zoos and Aquariums), and SEAZA (Southeast Asia Association of Zoos and Aquariums) provide some management and welfare guidelines for the various bird groups and species. Pet birds, however, are protected only by national anti-cruelty legislation that might affect feather plucking and other abnormal behaviors [89]. 

### 4.2. Legislation about Birds Taken from the Wild

Regulation of the legality of taking birds from wild can be directly covered by a domestic law, or might depend on reference in domestic law and enforcement of international codes, for example according to the EU Wild Birds Directive [90]. Some of the difficulties with the use of lists of protected species, e.g., by IUCN and CITES, are discussed in [91]. There are many differences between the species listed in the IUCN Red List or the CITES list, and the species given protection under national law. This might cause species on the CITES list to be legally bred and traded in the domestic market, a practice that encourages illegal smuggling. Most species of wild birds are not on a list of endangered species, and in many countries, there is nothing preventing them from being taken from the wild, with severe negative consequences for the welfare and survival of individuals, and for the population size of some species. Some of these issues are discussed by [4,91].

### 4.3. Legislation Wording

In every country in the world, new laws are needed in relation to keeping birds as pets and obtaining or breeding birds for sale to the pet trade. No bird should be taken from the wild and kept as a pet, or kept for sale to the pet trade. No bird should be taken from the wild and kept for display to the public or for breeding purposes, unless (i) the conditions of keeping and management meet all needs of the bird and there is a monitoring program ensuring good welfare, and (ii) it is part of an internationally recognized conservation program. If wild birds are found that are sick or injured, in most cases they should not be taken into captivity, as this makes them less likely to survive but in a few cases, captivity for a short period for veterinary treatment could be permitted. The breeding of birds for display to the public should utilize birds already in zoos and condition (i) above should be met. Whilst the evidence in this paper applies to birds, the same proposals could also apply to other wild caught animals, such as reptiles and mammals.

## 5. Conclusions

The trade in wild birds to be sold as pets is reducing biodiversity and the number of wild birds in the world, and this reduction is rapid for some species. Keys to the trade are the social aspects of the bird-keeping culture in Asia, which is the highest biodiversity region. It is also causing much suffering to large numbers of individual birds. The first issue when considering the sustainability of keeping birds as pets, concerns the origins of the birds. Many hundreds of species of birds kept as pets are taken from the wild. On grounds of welfare alone, mortality rates of 75–90% between capture and sale to the public make the practice unacceptable to most pet owners and most of the general public. The capture, holding, and transport procedures result in fear, disease, and other suffering for each of these birds. The taking of birds from the wild also has substantial negative impacts on conservation and biodiversity. Places where birds are bred for sale to the pet-owning public should not take birds from the wild, and should not be accepted as sources of any species of pet bird, if the survival and reproduction rate of birds of that species are significantly lower in the establishment than is possible in good wild conditions.

It is clear from the evidence presented here that there are many risks involved in much of the housing used for pet birds. Many current practices are unsustainable, as there are health problems and behavioral problems that occur as a result of keeping a bird in captivity. The most severe of these are the behavioral problems that are commonly documented in pet birds, resulting from not only the confined environment with inadequate mental stimulation, inadequate space, etc. but sometimes also from the rearing method that leads to behavioral problems. Given the high level of cognitive ability of parrots and other birds, it is extremely difficult to provide sufficient stimulation to keep them intellectually occupied when caged, especially when compared to the complexity of the wild environment. The lack of space for caged pet birds severely limits their behavior and result in muscle strength weakness. Birds should not be kept in captive conditions that result in stereotypies or self-mutilation, like feather-picking. Social birds, i.e., almost all pet birds, should not be kept isolated from other members of their species. All birds should be in aviaries that allow them to exercise, for example, at least five seconds of normal speed flight. How birds are being displayed and how enclosures are designed, should be considered carefully, especially the size and the vertical altitude of the cage, to allow birds to fulfill their needs to fly.

Furthermore, it can be concluded that there is a high prevalence of disease and injury in pet birds. Many of which stem from poor care by owners and inadequate routine maintenance, leading to poor living conditions for the birds. 

In relation to nutrition, there are many examples of nutritional disorders commonly found in pet birds. Pet bird owners should make sure that they are educated about proper nutrition and bird care.

It is not ethically right to keep the majority of the birds that are at present kept as pets, due to the origins of some species, and for others, the large number of nutritional, health, and behavioral problems that occur due to inadequate keeping conditions and incorrect husbandry by the bird owners. If birds are to be continued as a companion animal for people, then more effort should be made by pet shop owners and veterinarians to supply literature to prospective owners, highlighting not only the proper care for the species of bird but also its needs and requirements, so that bird owners can do their utmost to meet them in captivity. New laws are needed to prohibit taking birds from the wild and keeping birds in conditions that do not meet their needs, and some wording to achieve this is proposed.

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
