# Peer review of "The Sustainability of Keeping Birds as Pets: Should Any Be Kept?"

_animals, 2021, doi:10.3390/ani11020582_

Round 1

Reviewer 1 Report

Little things first.

On Line 44, change "the captive" to "a captive."  On line 322, "has can cause" doesn't make sense.  And on line 393, I think "does bird leave" should read "does the bird leave."

On to the bigger picture.

I think the title of the piece needs reconsideration. The manuscript as it stands doesn't directly answer the question of which birds should be kept in the sense that it identifies some named birds as being more "keepable" than others.  The argument the manuscript makes, quite rightly in my view, is that very few if any birds should be kept as pets/companions unless very substantial and much needed changes are made (a) in the pet trade in birds and (b) in the way people house and feed and care for their birds once they buy them and take them home. Indeed, the thrust of the manuscript is that it doesn't make much sense to ask "which (birds) should be kept?" The answer the authors supply is few, if any.  So, why is that question in the title of the piece?

There also needs to be more thought given to the manuscript's use of the word sustainable.  The argument advanced is essentially normative -- that people ought not to keep birds as pets because the way it's done now is morally and ethically suspect; that it is for the most part an unacceptable human practice.  But is the bird trade and the keeping of birds as pets sustainable in the sense that most people would think of when they see the word sustainable, namely will the world "run out" of birds if things keep going on the way they are now?  What is the impact as an empirical matter of keeping birds as pets, gauged either in relation to the total number of birds in the world and the viability of the environments in which they live or in relation to numbers and viability in particular places and species?  The manuscript says more than once that the mortality of birds entered into trade is appallingly high.  But it doesn't say anything as far as I can tell about the net impact of mortality and disease in the trade and in pet keeping on the world's populations of birds, except to say that in some (extreme?) cases the trade has been a direct cause of extinctions.  Overall, is the bird trade gradually or steadily reducing the number of birds in the world?  And what's the magnitude of that impact? Or is it the case that natural rates of reproduction, together with captive breeding, yield more than enough "new" birds to keep the trade in birds alive and profitable (and in that limited sense sustainable), even though it is morally reprehensible?

I think the other big picture question that the manuscript sidesteps but probably ought to address more directly has to do with law.  At the very end of the manuscript the authors say, almost as a throw away line, that new laws are needed to stop people taking birds from the wild and to make sure they keep them properly as pets. Easily said, very hard to accomplish. Given everything that's gone before in the manuscript, the assertion that new laws are needed is perhaps and in a gross sense an obvious thing to say.  But what exactly does it mean?  And, since the bird trade and the keeping of birds as pets/companions has been going on for a very long time, why hasn't law already addressed (at least some of the most egregious) risks and abuses the manuscript chronicles?  And if there are laws on the books -- CITES at the international level, for example, and national/state animal welfare statutes -- what's wrong with them and why don't they work?

Author Response

  1. The minor corrections suggested have been made.
  2. The title has been changed to: The sustainability of keeping birds as pets: should any be kept?
  3. Since sustainability is about all components and not just about “will the world run out of birds, or birds to keep?” we are trying to consider all of the components and “the magnitude of the impact”. We have improved the discussion to take account of this point.

  1. We have added text and a reference in relation to the impact of the pet trade on wild populations but have not carried out an extensive analysis of this.

  1. We have expanded the comments on law as suggested, have made them more precise and have added a section to do this.

Reviewer 2 Report

animals-1047058

GENERAL COMMENTS

This article discusses the welfare and mortality of birds in pet trading and keeping, and focuses on issues concerning mortality and ethics. These are important issues of concern to biologists, conservationists, veterinarians and the public globally.

The article is overall reasonably well written, although numerous sentences (indeed from the outset) are verbose and almost back-to-front (see ‘Specific comments’ below).

I think that the authors may also wish to refer to the current Animals special issue (Pets, People and policies’), in which can be found some proposed arguments and legal models  for ameliorating exotic pet trading and keeping problems (e.g. positive lists and bans), and they may find some of those items helpful to underscore/embolden their own recommendations.

https://www.mdpi.com/journal/animals/special_issues/Pets_People_Policies

In particular:

https://www.mdpi.com/2076-2615/10/12/2371

https://www.mdpi.com/2076-2615/10/12/2456

SPECIFIC COMMENTS

Lines 2-3

The title should have all primary words capitalised.

Lines 9-10

“A wide range of practices that result in many aspects of the keeping of birds as pets being unethical and unsustainable are described.”  

This would be better written as something like:

“We describe a wide range of unethical and unsustainable practices inherent to the trading and keeping of pet birds.”

I suggest that the authors go through the text with an eye for sentences like that one.

Lines 10-11, read:

“Wild-caught birds should not be sold to the public as pets, or to breeding establishments, because 75-90% of wild-caught birds die before the point of sale and taking birds from the wild has negative effects on biodiversity.”

I do not disagree with the authors. However, as written the sentence can be read as these are the only reasons for not selling or keeping birds. Other reasons are also as valid: IAS risk, zoonoses.

Relatedly, as read, it can be interpreted to imply that captive-bred birds are ok to sell and keep, which would be misleading, given that there are very many serious psycho-behavioural problems for many birds associated with captive-breeding and cage life in the home.

I should suggest that the authors widen their message so that it does not overlook the problems of pet birds generally (whether wild-caught or captive-bred origin), and relatedly widen their ‘not be sold/kept’ message to birds in general. It is perfectly reasonable to make the point that one would not keep a puppy or small dog in a bird cage in someone’s house, and that argument is at least as valid for birds whether wild-caught or captive-bred.

Line 19 reads:

‘At present’

Delete. This is a long-standing problem, not just the present.

Lines 20-23

Another example of a wordy, back-to-front sentence. I shall not keep raising them, but please go through and look for these.

Lines 24-26

Agreed, such recommendation for guidance is always sensible. However, there is now a significant body of work showing that even objective scientific/veterinary guidance is poorly followed by traders and private keepers.

e.g.

Abbott, S.L., Ni, F.C.Y. and Janda, J.M. (2012). Increase in extraintestinal infections caused by Salmonella entericasubspecies II-IV. Emerg. Infect. Dis., 18:(4);637–639. doi:10.3201/eid1804.111386.

Burman, O.H.P., Collins, L.M., Hoehfurtner, T., Whitehead, M. and Wilkinson, A. (2016)  Cold-blooded care: understanding reptile care and implications for their welfare. Testudo, 8,(3);83-86.

Howell, T.J., Bennett, P.C., 2017. Despite their best efforts, pet lizard owners in Victoria, Australia, are not fully compliant with lizard care guidelines and may not meet all lizard welfare needs. Journal of Veterinary Behavior 21, 26-37.

Howell, T.J., Warwick, C., Bennett, P.C., 2020. Self-reported snake management practices among owners in Victoria, Australia. Veterinary Record, 1-6.

Kohler R. Der ‘Schildkrötentest’: das neue Gesundheitsprojekt für engagierte Mitglieder. Elaphe. 2010;2:57-62.

Grant, R.A., Montrose, V.T., Wills, A.P. (2017) ExNOTic: Should we be keeping exotic pets? Animals, 7, 47.

Moorhouse, T.P., Balaskas, M., D'Cruze, N.C., Macdonald, D.W., 2017. Information could reduce consumer demand for exotic pets. Conservation Letters 10, 337-345.

Pees M, Mueller K, Mathes K, Korbel R, Seybold J, Lierz M, et al. Evaluation of

Whitehead, M.L., 2018. Factors contributing to poor welfare of pet reptiles. Testudo 8, 47-61.

Whitehead, M.L., Vaughan-Jones, C., 2015. Suitability of species kept as pets. Veterinary Record 177, 573.

Arguably, therefore, greater emphasis is needed on the authors’ ‘new laws’ recommendation, because it will indeed require major legal control of the pet trade to achieve good welfare results, and education has a poor record.

Lines 26-28 read:

“New laws are needed to prohibit taking birds from the wild and keeping birds in conditions that do not meet their needs.”

This sentence is repeated at lines 29-30.

Lines 45 -49

This is a very good point, and maybe even poultry could add to this example. Broiler chickens are multigenerational captive-bred, and also specifically bred for captivity. Yet, they do not adapt to cage life, suffer greatly as a result, and when released into naturalistic conditions rapidly revert to hard-wired behavioural norms - which they are probably trying and failing to express in cages. Thus, pet birds in captivity certainly cannot be considered ‘domesticated’ or adapted to cage life, regardless of origin.

Line 83 (2.3) I note that the authors are starting to address the IAS/other issues here, which is good, but I still suggest that these other issues are also mentioned in the Synopsis/Abstract.

Lines 1810186

This sounds like the authors are condoning wing-clipping. Surely, this is one ‘evil’ to guard against another, and would require significant ethical justification that is not clear or justified

from the authors’ text.

Finally, I reiterate that to keep their article consistent, I think the authors would need consider expand their message re new laws and prohibitions to include birds generally, given the inherent problematic nature of trading and keeping these animals.

Author Response

  1. In the new section on Laws, reference has been made to Toland et al (2020).

In the Introduction reference has been made to D’Cruze et al (2020) and three other papers.

2-3 capitalised

9-10 sentence changed as suggested

10-11 sentence changed to: “Wild-caught birds should not be sold to the public as pets, or to breeding establishments, for several reasons, including that 75-90% of wild-caught birds die before the point of sale and taking birds from the wild has negative effects on biodiversity.”

Sentence added after … the risk of disease “Unless these changes are made, the keeping of birds should not be permitted.”

19 At present removed. Added at end of first sentence “which are currently widespread.”

20-23 sentence changed to: ‘Some pet birds were taken from the wild or bred in poor conditions while others have nutritional, health and behavioural problems resulting from inadequate keeping conditions and incorrect husbandry by the bird owners. As a consequence, it is not ethically right to keep the majority of the birds that are at present kept as pets.’

24-26 add after “in captivity” ‘Owners do not comply with laws requiring duty of care unless they obtain and act on such information and also knowledge of how to provide good nutrition and minimise the risk of disease.’

Some of these references are now cited in the text. e.g. Abbott et al 2012, Moorhouse et al, 2017 but we have said little about reptiles so do not include those references.

26-28 and 28-29  Duplicated sentence removed. “New laws are needed to prohibit taking birds from the wild and keeping birds in conditions that do not meet their needs. Owners should have knowledge of how to provide good nutrition and minimise the risk of disease.”

45-49 Thanks for your comment. We think it would be confusing to refer to commercial poultry here.

83 references added to clarify and abstract improved.

181-186 line 181 changed to: “However, wing-clipping should be avoided, except in rare situations where it improves bird welfare, and the process needs to be carried…”

“Finally”  We have added a new section relating to laws.

Round 2

Reviewer 1 Report

The revisions made by the authors adequately address the comments made previously.

HOWEVER, the revisions have introduced a number of additional minor errors.  And these now need to be corrected.

Specifically:

line 5 -- Insert a period and a space after the word Taiwan in the leade author's address.

line 33 -- Remove the period after the word questioned

line 410 -- The word Bird at the start of this paragraph should read Birds (plural)

lines 411-412 -- Revise the sentence beginning "The laws vary..." so that it reads: "The laws vary but may cover factors affecting welfare, such as pinioning, flight restraint or cage size."  As it's now written, the sentence doesn't make sense.

line 418 -- Insert a definite article between birds and from so that it reads "taking birds from the wild..."

line 419 -- Change "reference in domestic law and enforcement" so that it reads "references in domestic law to the enforcement of international codes."

line 436 -- Remove the comma between the words cases and captivity

lines 491-618 -- The References section needs careful copy editing to re-format the entire section and re-number all the items cited.

Author Response

 All small changes and number the references has been edited according to Reviewer's suggestions.  Thank you, Shawn
